# Object color knowledge representation occurs in the macaque brain despite the absence of a developed language system

**Minghui Zhao**[1,2], **Yumeng Xin**[1,2], **Haoyun Deng**[1,2], **Zhentao Zuo**[1,2,3], **Xiaoying Wang**[4,5]\*, **Yanchao Bi**[4,5,6,7]\*, **Ning Liu**[1,2]\*

1 State Key Laboratory of Brain and Cognitive Science, Institute of Biophysics, Chinese Academy of Sciences, Beijing, China, 2 College of Life Sciences, University of Chinese Academy of Sciences, Beijing, China, 3 Sino-Danish College, University of Chinese Academy of Sciences, Beijing, China, 4 State Key Laboratory of Cognitive Neuroscience and Learning, Beijing Normal University, Beijing, China, 5 IDG/McGovern Institute for Brain Research, Beijing Normal University, Beijing, China, 6 Beijing Key Laboratory of Brain Imaging and Connectomics, Beijing Normal University, Beijing, China, 7 Chinese Institute for Brain Research, Beijing, China

\* wangxiaoying@bnu.edu.cn (XW); ybi@bnu.edu.cn (YB); liuning@ibp.ac.cn (NL)

**Data Availability Statement:** All relevant data are within the paper and its Supporting Information files. All data files are available from the Zenodo

## Abstract

Animals guide their behaviors through internal representations of the world in the brain. We aimed to understand how the macaque brain stores such general world knowledge, focusing on object color knowledge. Three functional magnetic resonance imaging (fMRI) experiments were conducted in macaque monkeys: viewing chromatic and achromatic gratings, viewing grayscale images of their familiar fruits and vegetables (e.g., grayscale strawberry), and viewing true- and false-colored objects (e.g., red strawberry and green strawberry). We observed robust object knowledge representations in the color patches, especially the one located around TEO: the activity patterns could classify grayscale pictures of objects based on their memory color and response patterns in these regions could translate between chromatic grating viewing and grayscale object viewing (e.g., red grating—grayscale images of strawberry), such that classifiers trained by viewing chromatic gratings could successfully classify grayscale object images according to their memory colors. Our results showed direct positive evidence of object color memory in macaque monkeys. These results indicate the perceptually grounded knowledge representation as a conservative memory mechanism and open a new avenue to study this particular (semantic) memory representation with macaque models.

## Introduction

Our brain constructs rich internal representations of the external world—the color of a strawberry, the shape of predators, the functions of social systems, etc. Such (explicit or implicit) general world "knowledge" is referred to as semantic memory, in contrast to episodic memory, which is more about time- and space-sensitive experiences. Semantic memory supports a remarkably wide range of essential everyday behaviors, from recognizing an object to using

database (URLs:https://zenodo.org/records/13739051).

**Funding:** This study was supported by the Ministry of Science and Technology of China (https://en.most.gov.cn/) STI2030-Major Projects (2021ZD0200200 to N.L., 2021ZD0204200 to M,Z, 2021ZD0204100 to Y.B.), the Natural Science Foundation of China (https://www.nsfc.gov.cn/english/site_1/index.html, 31925020 to Y.B., 32071050 to X.W.), the Fundamental Research Funds for the Central Universities (http://en.moe.gov.cn/, to Y.B. and X.W.), and the University Synergy Innovation Program of Anhui Province (http://jyt.ah.gov.cn/, GXXT-2021-002 to Z.Z., GXXT-2022-029 to Z.Z.), and the Youth Innovation Promotion Association CAS (http://www.yicas.cn/, 2021091, YSBR-068 to Z.Z.). The funders had no role in study design, data collection and analysis, decision to publish, or preparation of the manuscript.

**Competing interests:** The authors have declared that no competing interests exist.

**Abbreviations:** ATL, anterior temporal lobe; CAS, Chinese Academy of Sciences; dATL, dorsal anterior temporal lobe; FDR, false discovery rate; fMRI, functional magnetic resonance imaging; GLMM, generalized linear mixed model; IT, inferotemporal; LDA, linear discriminant analysis; MEG, magnetoencephalography; MION, monocrystalline iron oxide nanocolloid; MR, magnetic resonance; MVPA, multivoxel pattern analysis; PR, perirhinal cortex; ROI, region of interest; SNR, signal-to-noise ratio; TP, temporal pole; VPC, visual paired comparison.

complex tools and language [1,2]. How semantic memory (i.e., general knowledge) is represented in the brain has been studied extensively in humans, yet it remains unclear where the evolutionary origin of such knowledge representations is.

For human semantic memory research, mounting brain imaging and neuropsychological evidence highlight a distributed cortical network encompassing temporal, parietal, and frontal regions [3]. One consensus is that such a distributed pattern contains various (higher-order) sensorimotor brain systems, reflecting at least in part knowledge of attribute "grounded" in various sensorimotor experiences and their interactions [2,4–7]. A classic example is object color knowledge. Retrieving color knowledge of an object or reading words denoting objects with diagnostic color property (e.g., strawberry) activates ventral occipitotemporal regions close to/overlapping with color patches in the human brain. This territory's intrinsic activity strength and connectivity strength correlate with the retrieval efficiency of object color knowledge, and lesions encompassing this territory are associated with specific object color knowledge impairment [8–15]. The knowledge about specific physical properties perceived by various sensory modalities (e.g., color, shape, taste, haptic) is gradually abstracted and bounded into more integrative object representations (e.g., strawberry versus banana) in the anterior temporal lobe (ATL) [7,16–21]. Compelling as this framework is, the effect of language is intrinsically confounded with sensorimotor experiences in shaping and probing human semantic memory [6,22–25]. The study of nonhuman primates is critical in examining the conservativeness of the neural mechanisms underlying semantic memory (i.e., general knowledge) representation without a developed language system and offering the spectrum of methodologies to potentially study such neural mechanisms at scales not possible with human subjects.

However, research on the neural basis of "semantic-like" memory in nonhuman primates is scarce, with the majority of studies focusing on the acquisition of artificial associations that are abstracted away from specific perceptual properties rather than natural "world-knowledge" memory representations about specific perceptual properties themselves [26–32]. Another line of research has been focused on familiarity, a broad sign of past information registered in the brain [33,34]. Both lines of studies have indicated the involvement of ATL, such as the temporal pole (TP) and perirhinal cortex (PR), in the representation of integrated object memory. What is critical yet unknown is whether the perceptual system of nonhuman primates also contributes to object knowledge representation by directly supporting the memory about corresponding object attributes based on perceptual experiences. The ventral visual pathway of the macaque brain has been particularly well studied for its specialized response profiles to various visual properties [35]. A set of "color patches" has been identified within it, showing selectivity for chromatic compared to achromatic gratings [36,37]. Do these color patches support object color memory, which can be retrieved without color input, like previously found in human brains?

Here, 3 fMRI experiments with rhesus monkeys were carried out: chromatic and achromatic grating viewing (Exp 1), grayscale object viewing (Exp 2), and true- and false-colored object viewing (Exp 3). The 3 experiments were combined to investigate the stored object color knowledge in the macaque brain. In addition, to directly investigate the existence of object color knowledge in tested animals in the present study, we performed a behavioral experiment using the same stimuli as those in Exp 3.

## Results

To investigate the neural substrates underlying the object color knowledge in macaques, we conducted 3 fMRI experiments in 3 rhesus monkeys using stimuli that they are familiar with

through life experience. Upon their arrival at the laboratory, their diet included selected fruits and vegetables with 3 diagnostic colors (Red: strawberry, watermelon; Green: cabbage, kiwi; Yellow: banana, corn). The 3 fMRI experiments probing color perception and object color knowledge included the following (Fig 1): (1) Chromatic and achromatic grating viewing (Exp 1): Macaques viewed luminance-matched chromatic (i.e., red, green, yellow) and achromatic (i.e., 25%, 50%, 75% luminance contrast) gratings, the contrast of which facilitated color patch localization; (2) grayscale object viewing (Exp 2): Macaques viewed grayscale photos of fruits and vegetables with 3 diagnostic colors (Red: strawberry, watermelon slice; Green: cabbage, kiwi slice; Yellow: banana, corn), and we tested whether brain responses to grayscale images could accurately encode objects' typical colors (i.e., reflecting memory representation); (3) true- and false-colored object viewing (Exp 3): Macaques viewed photos of correctly colored objects (i.e., red strawberry, red watermelon slice, green cabbage, green kiwi slice) and their color-swapped false counterparts (i.e., red-colored cabbage and kiwi slice; green-colored strawberry and watermelon slice), and the true/false color univariate differentiation and multivariate decoding accuracy from brain responses were examined (i.e., testing if a region could distinguish visual stimuli that correspond to real natural objects from those that do not). To avoid interference between experiments caused by the potential learning effect, we conducted the experiments in the following order: Exp 2, Exp 1, and Exp 3. Furthermore, we conducted a free-viewing visual paired comparison (VPC) task to investigate subjects' visual preference for true- and false-colored object stimuli used in Exp 3 (see S1 Text and S1A Fig).

## Color patches localization

We first localized color patches along the ventral visual pathway by contrasting chromatic and achromatic gratings (Fig 1B, Exp 1). Following the previous procedure used for color patch identification [37,38], the effect of luminance was controlled by referencing the activation in MT, which is specialized to process moving stimuli and responds less strongly to moving chromatic gratings if the foreground and background within the chromatic gratings are equiluminant. We identified the chromatic grating that induced the weakest activation in MT for each monkey, as well as the corresponding achromatic grating that evoked comparable activation with the selected chromatic grating in V1. We then contrasted the activation of the selected chromatic grating and the corresponding achromatic grating, and located a series of color

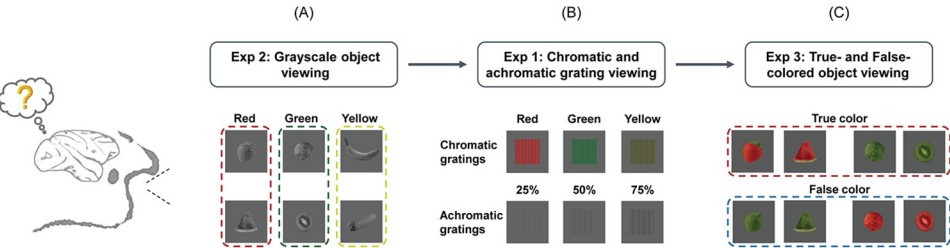

**Fig 1. Experimental procedures.** Three experiments were conducted and combined to investigate the neural basis for object color and related integrative object knowledge. (A) Examples of stimuli in the grayscale object viewing experiment (Exp 2): 6 types of color-diagnostic grayscale objects from 3 color categories (Red: strawberry, watermelon; Green: cabbage, kiwi; Yellow: banana, corn). (B) Examples of stimuli in the chromatic and achromatic grating viewing experiment (Exp 1): 3 equiluminant chromatic gratings (i.e., red, green, and yellow) and 3 achromatic gratings with 25%, 50%, and 75% luminance contrast that had equal mean luminance with chromatic gratings. (C) Examples of stimuli used in the true- and false-colored objects viewing experiment (Exp 3). Only red and green color categories (Red: strawberry, watermelon; Green: cabbage, kiwi) and their color-swapped false items were included in Exp 3. Three fMRI experiments were performed in the order of Exp 2, Exp 1, and Exp 3 to avoid interference between experiments caused by learning. fMRI, functional magnetic resonance imaging.

patches (chromatic > achromatic) in all 3 animals along the ventral visual pathway, including the color patches around V4d (V4d_c), V4v (V4v_c), TEO (TEO_c), TEpd (TEpd_c), TEad (TEad_c), TEav (TEav_c), and the boundary between areas TEa and IPa (TEa_c) (Fig 2A). Note that, to signify the locations of the color patches, we named the color patches based on their anatomical locations by appending "_c" at the end of the brain regions where the color bias clusters were located. Please refer to S1 Table for the correspondence between the names in the present study and the corresponding names in previous studies [36,37]. Region of interest (ROI) analyses contrasting the activation of the unselected chromatic and achromatic gratings confirmed the color sensitivity in these color patches [see S2A Fig for the group generalized linear mixed model (GLMM) results with false discovery rate (FDR) correction (aka q-value); V4d_c: $F_{(1,156)}$ = 164.141, q < 0.001, Cohen's d = 2.052; V4v_c: $F_{(1,155)}$ = 155.161, q < 0.001, Cohen's d = 2.001; TEO_c: $F_{(1,156)}$ = 143.046, q < 0.001, Cohen's d = 1.915; TEpd_c: $F_{(1,156)}$ = 81.590, q < 0.001, Cohen's d = 1.446; TEad_c: $F_{(1,156)}$ = 61.739, q < 0.001, Cohen's d = 1.258; TEav_c: $F_{(1,156)}$ = 6.603, q = 0.011, Cohen's d = 0.412; TEa_c: $F_{(1,156)}$ = 9.556, q = 0.002, Cohen's d = 0.495; two-tailed]. All of these color patches could successfully discriminate the 3 colors in the multivoxel pattern analysis (MVPA) [S2B Fig; V4d_c: $F_{(1,156)}$ = 205.489, q < 0.001; V4v_c: $F_{(1,156)}$ = 103.752, q < 0.001; TEO_c: $F_{(1,156)}$ = 209.230, q < 0.001; TEpd_c: $F_{(1,156)}$ = 63.425, q < 0.001; TEad_c: $F_{(1,156)}$ = 10.894, q < 0.001; TEav_c: $F_{(1,156)}$ = 8.245, q = 0.002; TEa_c: $F_{(1,156)}$ = 43.701, q < 0.001; one-tailed; see the results of each monkey in S3 Fig].

## Object color memory representation in the color patches

**Classification of grayscale objects with different memory colors.** We compared averaged responses to 3 categories of grayscale objects. No significant differences were found in any of the defined color patches (S4A Fig). To investigate whether color patches can encode object color knowledge in the absence of color inputs, we started by examining whether the grayscale objects with different memory colors (e.g., strawberry–kiwi) would elicit distinct response patterns, whereas those with the same memory color (e.g., strawberry–watermelon) would evoke similar response patterns in color patches.

Firstly, we trained a linear discriminant analysis (LDA) classifier [39] with a half set of the color-diagnostic objects (e.g., strawberry, cabbage, and banana) and tested it on the other half set of objects (e.g., watermelon slice, kiwi slice, and corn). As shown in S5A Fig, response patterns in color patches V4d_c, V4v_c, TEO_c, TEad_c, and TEa_c could successfully distinguish 3 categories of grayscale objects with different memory colors.

Note that previous studies have reported that color patches can also represent shape information to some extent [36]. Therefore, the above-mentioned findings in color patches may be driven by memory colors and/or shape information. To ensure the exclusion of confounding effects related to object shape, we excluded objects with yellow memory color (i.e., banana and corn) due to their relatively distinct shapes (elongated) compared to the other objects [S6A Fig; significant shape difference from the red category: $F_{(1,118)}$ = 33.396, $p$ < 0.001; from the green category: $F_{(1,118)}$ = 33.639, $p$ < 0.001], and then reran the above analyses. Note that no significant shape differences were found between grayscale objects with green and red memory colors [S6A Fig, $F_{(1,118)}$ = 1.860, $p$ = 0.175]. Four times cross-validation analyses (Fig 2B) were conducted and then averaged to yield the final decoding accuracy. This more rigorous analysis showed that response patterns in color patches V4d_c, TEO_c, and TEad_c could successfully distinguish grayscale objects with green and red memory colors [Fig 2C; V4d_c: $F_{(1,176)}$ = 148.933, q < 0.001; V4v_c: $F_{(1,176)}$ = 2.136, q = 0.102; TEO_c: $F_{(1,176)}$ = 28.520, q < 0.001; TEpd_c: $F_{(1,176)}$ = 6.118, q = 0.993; TEad_c: $F_{(1,176)}$ = 5.371, q = 0.025; TEav_c: $F_{(1,176)}$ = 5.589,

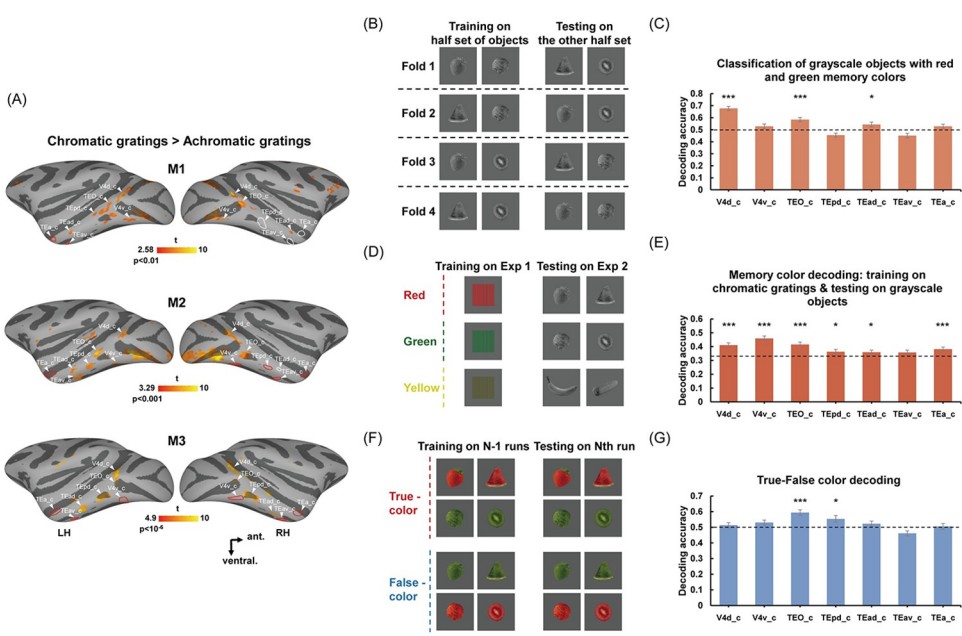

**Fig 2. Representations of object color memory in the color patches.** (A) Color patches (chromatic versus achromatic grating) from Exp 1 are shown on lateral views of the template inflated surface. Note that relatively higher thresholds ($p < 0.01$ for M1, $p < 0.001$ for M2, and $p < 10^{-6}$ for M3) were applied to depict the locations of color patches without confluent activation. Red solid lines indicate that color patches ($p < 0.05$ uncorrected) could not be presented at the thresholds set for A. For M1 and M2, the achromatic grating was adjusted to the next lower level (i.e., from 50% to 25%) to localize the color patches with weak color bias in the right hemisphere [i.e., TEpd_c (CLc), TEav_c (AVc), and TEa_c (AFc) in M1, and TEad_c (ALc) in M2] at a threshold of $p < 0.05$ (uncorrected). These color patches are marked with solid white lines. (B, C) The illustration and results of classification of grayscale objects with red and green memory colors: training the classifier to distinguish a half set of the red and green color-diagnostic grayscale objects and testing on the other half. Successful memory color decoding was found in V4d_c, TEO_c, and TEad_c. (D, E) The illustration and results of memory color decoding based on chromatic gratings training: training the classifier to distinguish among 3 chromatic gratings on Exp 1 and then testing on 3 categories of grayscale objects in Exp 2. Successful memory color decoding was found in V4d_c, V4v_c, TEO_c, TEpd_c, TEad_c, and TEa_c. (F, G) The illustration and results of true-false color decoding: training on true- and false-colored objects in N-1 runs and testing on the left-out run. Significant decoding of true-false color was done only in TEO_c and TEpd_c. Bars display mean values +/− SEM. Black asterisks indicate a significant difference from the chance level (0.333 in E, 0.5 in C, and G, indicated by the dashed lines); *q < 0.05, **q < 0.01, ***q < 0.001. The data underlying this figure are available in S1 Data and https://zenodo.org/records/13739051.

q = 0.993; TEa_c: $F_{(1,176)} = 2.464$, q = 0.102; one-tailed]. At the individual level, significant results were found in V4d_c and TEO_c for all 3 subjects (see S7A–S7C Fig for the results of each monkey). Whole-brain searchlight results also showed that areas around the posterior color patches could successfully discriminate between objects based on their memory colors in all 3 subjects (see S7D–S7F Fig for the results of individual monkeys).

To further exclude the potential shape confound on memory color decoding, we examined the classification results for cases where shape could not help the memory color classification (for details, see S5B Fig, as marked with red frames). As shown in S5C Fig, even in this more rigorous analysis, we again observed successful differentiation between grayscale objects with different memory colors in V4d_c, TEO_c, and TEad_c [V4d_c: $F_{(1,176)} = 116.955$, q < 0.001; V4v_c: $F_{(1,176)} = 1.124$, q = 0.203; TEO_c: $F_{(1,176)} = 11.836$, q = 0.001; TEpd_c: $F_{(1,176)} = 2.725$, q = 0.088; TEad_c: $F_{(1,176)} = 7.728$, q = 0.007; TEav_c: $F_{(1,176)} = 4.007$, q = 0.977; TEa_c: $F_{(1,176)} = 0.473$, q = 0.287; one-tailed].

**Decoding memory colors of grayscale objects based on responses to chromatic gratings.** To more directly examine whether the object color knowledge representation aligns

with actual color perceptual experience and to further eliminate the shape confound, we trained an LDA classifier with the brain activity patterns elicited by the chromatic gratings (i.e., red, green, yellow) in the chromatic and achromatic grating viewing experiment and tested it on the activity patterns evoked by the grayscale objects with different memory colors in the grayscale object viewing experiment (Red: strawberry, watermelon slice; Green: cabbage, kiwi slice; Yellow: banana, corn; Fig 2D). Since the classifier was specifically trained to discriminate among chromatic gratings with only differences in color but not shape or other information, this approach has been shown to effectively address concerns about potential confounds and was widely used in many previous human studies [10,40,41]. Classification accuracies across 3 monkeys in the V4d_c, V4v_c, TEO_c, TEpd_c, TEad_c, and TEa_c were significantly higher than the chance level (0.333) [Fig 2E; V4d_c: $F_{(1,176)}$ = 24.816, q < 0.001; V4v_c: $F_{(1,176)}$ = 65.443, q < 0.001; TEO_c: $F_{(1,176)}$ = 27.569, q < 0.001; TEpd_c: $F_{(1,176)}$ = 3.485, q = 0.045; TEad_c: $F_{(1,176)}$ = 3.176, q = 0.045; TEav_c: $F_{(1,176)}$ = 2.157, q = 0.072; TEa_c: $F_{(1,176)}$ = 12.256, q < 0.001; one-tailed]. Especially, in V4v_c and TEO_c, we consistently observed significant or marginally significant results across all 3 subjects (see the results of each monkey in S8A–S8C Fig). Whole-brain searchlight analyses in all 3 monkeys also confirmed that areas whose neural patterns successfully translate between chromatic gratings and grayscale objects with corresponding memory colors were mainly distributed in the color patches and their surrounding regions (S8D–S8F Fig).

As shown in S7D–S7F and S8D–S8F Figs, the searchlight results did reveal clusters in the primary visual cortex (V1) that could significantly encode memory color. However, it is worth noting that the location of these clusters was not consistent across 3 monkeys and across the memory color decoding approaches. Further analyses based on the defined ROI in V1 (the top 50 voxels with the strongest averaged responses to 6 grayscale objects) showed that V1 could not classify grayscale objects with red and green memory colors [S9A Fig: $F_{(1,176)}$ = 0.031, p = 0.431; one-tailed] but could successfully encode memory color when trained on chromatic gratings [S9B Fig: $F_{(1,176)}$ = 3.484, p = 0.032; one-tailed].

**Differentiation between true- and false-colored objects.** We further corroborated the findings of object color memory representation in the color patches with a true- and false-colored object viewing experiment (e.g., red strawberry versus green strawberry, Fig 1C, Exp 3). The false-colored object images were created by swapping the colors among the true-colored object items. Thus, the true- and false-colored objects were perfectly matched in terms of shapes and colors. The only difference between these 2 conditions was whether the combination of color and shape information matched the long-term memory representations of these objects obtained from daily experience. We first compared the activation strengths to true- and false-colored objects. Among the color patches, no regions exhibited significant differences in activation strength between these 2 conditions (S10 Fig; see S2 Table for statistical results of each region). MVPA decoding between true- and false-colored objects revealed that the activation patterns in color patches TEO_c and TEpd_c could differentiate between the 2 types of object images [Fig 2F and 2G; V4d_c: $F_{(1,128)}$ = 0.748, q = 0.272; V4v_c: $F_{(1,128)}$ = 4.176, q = 0.050; TEO_c: $F_{(1,128)}$ = 34.153, q < 0.001; TEpd_c: $F_{(1,128)}$ = 7.123, q = 0.015; TEad_c: $F_{(1,128)}$ = 1.907, q = 0.149; TEav_c: $F_{(1,128)}$ = 5.202, q = 0.988; TEa_c: $F_{(1,128)}$ = 0.101, q = 0.438; one-tailed]. The classification accuracy for all 3 monkeys surpassed the chance level, with significant results found in 2 of the 3 subjects (see S11A–S11C Fig for the results of individual monkeys). Similar results were obtained in the whole-brain searchlight analyses: significant true-false encoding regions were distributed mainly in the dorsal inferotemporal (IT) cortex, particularly surrounding the TEO_c and TEpd_c (S11D–S11F Fig).

Taken together, we consistently observed the representation of object color memory in the posterior color perceptual regions (especially in TEO_c) across multiple experiments and decoding analyses.

Note that there was some inter-hemispheric and inter-animal variability in the localization of defined color patches, as discussed in the previous study [37]. To ensure greater consistency in the locations of the color patches across different monkeys, we also conducted analyses on color patches defined on the averaged activation map across 3 subjects (see S2 Text and S12A Fig). Although the group-defined ROIs might not correspond to the most color biased voxels for each individual subject, we again observed similar results, especially for the color patch around TEO (S12B–S12F and S4B Figs).

## Object color memory representation in the ATL

**Experience-sensitive univariate differentiation between true- versus false-colored object images in TP.**   Previous studies have revealed the critical role of perirhinal circuits in the ATL in object-associative memory retrieval [26,42]. Therefore, we investigated 2 key nodes in this circuit: TP and PR. As the true- and false-colored object viewing experiment (e.g., red strawberry versus green strawberry, Fig 1C, Exp 3) was conceptually similar to previous associative memory studies utilizing object familiarity [26,33], we first examined the object memory representation in TP and PR in this experiment. When comparing responses to true- versus false-colored objects across all the sessions, we observed consistent bilateral activation strength differences between the 2 conditions in TP (Fig 3A, $p < 0.05$, uncorrected) and PR but only at a lenient threshold (S13A Fig) in all 3 monkeys. In addition, in line with previous studies in humans [43], the whole-brain analysis revealed widespread regions exhibiting stronger responses to false-colored stimuli compared to true-colored stimuli (S14 Fig).

Note that previous research has emphasized the crucial role of the ATL in signaling object familiarity, with neural responses being modulated by repeated stimulus exposure [44–47]. To investigate such a role of TP, we divided sessions into the first half and second half. Subsequently, we examined responses to true-colored versus false-colored objects within each half of the data. Remarkably, a dynamic and short-term "familiarization" process was observed in TP across all 3 monkeys. Specifically, TP exhibited significantly stronger responses to true-color objects compared to false-color objects during the first half of the sessions (Fig 3B). However, this differential response was no longer evident during the second half of the sessions (Fig 3C). Subsequently, we conducted a conjunction analysis that combined the true-color versus false-color contrast with the interaction effect between Period (first half versus second half of sessions) and true-false (S15A Fig). We did observe significant clusters in TP across all 3 subjects ($p < 0.01$, S15A Fig). That is, the whole-brain analyses consistently identified a significant short-term learning effect in TP for all 3 subjects.

ROI analyses were also conducted in TP. To avoid the possible circularity, we used a one-half set of the true- and false-colored objects to define the true-false ROI in TP and then performed analyses on the other half set of the stimuli. Similar results were found (S15B–S15G, S16A, and S16B Figs and S3 Table), albeit to a lesser significance due to the halved sample size, as required to avoid double dipping.

Different from TP, PR did not show any consistent short-term "familiarization" trend across 3 monkeys (S13B and S13C Fig, the whole-brain analysis; S13D, S13E, and S17 Figs, the ROI analysis).

We also examined whether the activation pattern in TP coded true- and false-colored objects differently. The MVPA showed that TP could not significantly differentiate between these 2 conditions either when combining all sessions [Fig 3D: $F_{(1,128)} = 1.976$, q = 0.162; one-

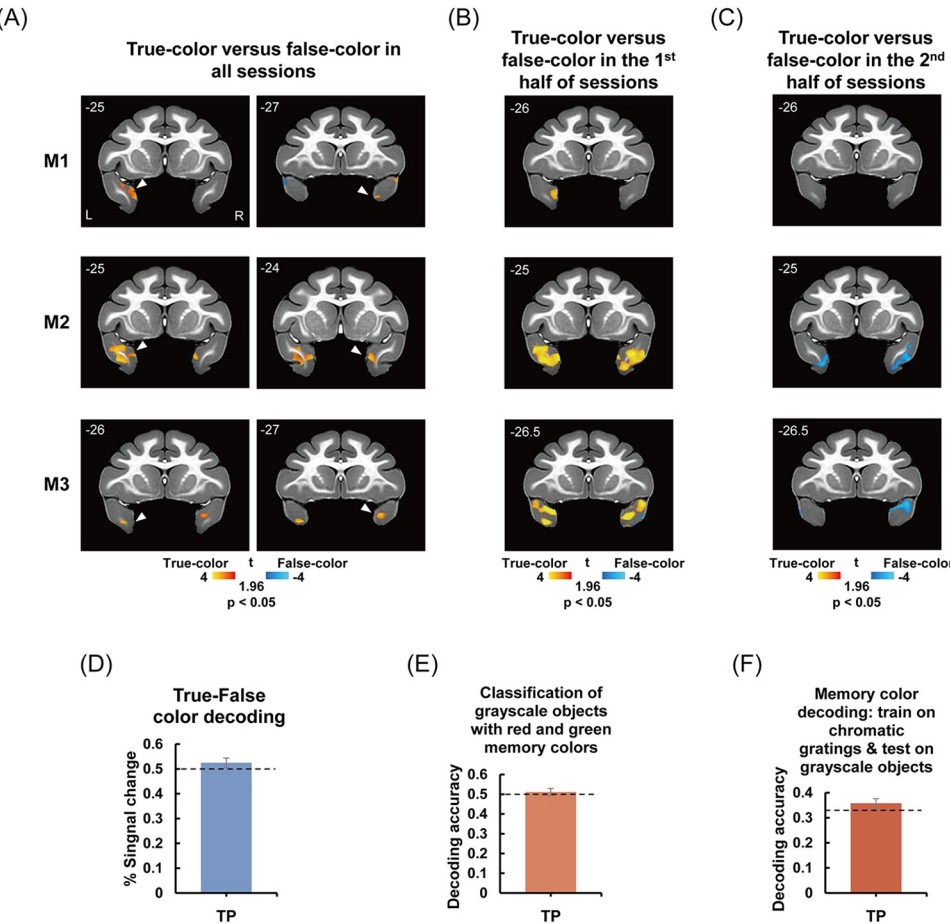

**Fig 3. Representations of object color memory in TP.** (A–C) The true-color versus false-color in all sessions (A), the first part of sessions (B), and the second part of sessions (C) in TP are shown in the coronal slices for each of the 3 subjects (M1 to M3) at $p < 0.05$, respectively. Each slice's anterior/posterior position is indicated on the top left corner (mm relative to the interaural canal). (D) True-false color decoding accuracy when combining all the sessions in TP. (E) Classification of grayscale objects with red and green memory colors in TP. (F) Memory color decoding accuracy based on chromatic gratings training in TP. Bars display mean values +/− SEM. The dashed lines indicate chance levels (0.5 in D and E, 0.333 in F). The data underlying this figure are available in S1 Data and https://zenodo.org/records/13739051. TP, temporal pole.

tailed; see S18A–S18C Fig for the results of individual monkeys] or when the first and second halves were analyzed separately [S19A Fig; first half (all sessions): $F_{(1,66)} = 0.020$, $q = 0.556$; second half (all sessions): $F_{(1,60)} = 0.012$, $q = 0.543$; one-tailed; see S18D–S18F Fig for the results of individual monkeys].

We conducted the same decoding analyses in PR as those performed in TP and did not find significant results (S18 and S19 Figs).

To check whether the above-mentioned learning effect also existed in the color patches, we conducted the same analyses in the color patches. None of these regions showed significant modulation by the short-term familiarization of false-colored objects in either the univariate contrast analyses (S20A Fig and S4 Table) or the MVPA [S20B Fig; V4d_c: $F_{(1,63)} = 1.634$, $q = 0.401$; V4v_c: $F_{(1,63)} = 0.009$, $q = 0.925$; TEO_c: $F_{(1,63)} = 2.343$, $q = 0.401$; TEpd_c: $F_{(1,63)} = 1.777$, $q = 0.401$; TEad_c: $F_{(1,63)} = 0.635$, $q = 0.600$; TEav_c: $F_{(1,63)} = 1.476$, $q = 0.401$; TEa_c: $F_{(1,63)} = −0.196$, $q = 0.770$; two-tailed; see S21 Fig for the results of individual monkeys]. Note that the whole-brain analysis revealed that only TP showed a significant interaction effect

between Period (first half versus second half of sessions) and true-false at the group level (S16C Fig).

**Lack of specific coding of object memory color in TP.** Unsurprisingly, there were no significant differences in responses to 3 categories of grayscale objects in TP (S4C Fig). We did not observe successful encoding of object memory color of grayscale objects in TP in the whole-brain searchlight results (see S7D–S7F and S8D–S8F Figs). To conservatively test whether TP also encoded the specific object color information, we examined memory color encoding of grayscale objects in the defined ROI in TP. Still, the object color information could not be significantly readout (classification of grayscale objects with red and green memory colors: see Fig 3E for the averaged results across 3 monkeys [$F_{(1,176)} = 0.401$, q = 0.527; one-tailed], see S22A Fig for the results of individual monkeys; memory color decoding based on chromatic gratings training: see Fig 3F for the averaged results across three monkeys [$F_{(1,176)} = 1.710$, q = 0.161; one-tailed], see S22B Fig for the results of individual monkeys). These results suggested that color information represented in TP was no longer specifically coded as a separate object feature.

Again, no significant results were found in PR in the same decoding analyses as those performed in TP (S22 Fig).

## Differences in behavioral responses to true- and false-colored object images

To direct examine the existence of object color knowledge in the tested animals in the present study, we conducted one behavioral experiment. Two subjects (M2 and M3 but not M1 due to health concerns) were trained to perform a VPC task [48–50], in which a true-colored object image from the fMRI experiments was paired with the corresponding false-colored image (S1A Fig). Specifically, we tested 2 indicators: the proportion of fixation time and the proportion of first fixation [51]. The results from both indicators showed that monkeys had a stronger preference for true-colored stimuli than false-colored ones [S1B and S1C Fig; the proportion of fixation time: $F_{(1,84)} = 28.607$, $p < 0.001$, Cohen's d = 1.167; the proportion of first fixation: $F_{(1,84)} = 33.688$, $p < 0.001$, Cohen's d = 1.267; two-tailed], which indicated that monkeys are behaviorally capable of systematically distinguishing between true and false-colored stimuli, confirming the presence of object color knowledge.

## Discussion

We aimed to discover how the macaque brain stores general information about the world, testing specifically the color knowledge of objects they are familiar with. The tested macaques showed a visual preference for true-colored objects, providing direct positive behavioral evidence of object color memory in macaques. Across multiple fMRI analyses, we found indications of object color knowledge representation in the color patches. Brain activity patterns in color patches, especially TEO_c, could classify grayscale object images based on their memory colors, with their activation patterns being transferrable among grayscale objects within the same diagnostic color category (e.g., grayscale images of watermelon—grayscale images of strawberry). Notably, activity patterns of these color patches could translate between chromatic grating viewing and grayscale object viewing (e.g., red grating—grayscale images of strawberry), such that classifiers trained by viewing chromatic gratings could successfully classify grayscale object images according to their diagnostic colors. Furthermore, the activity patterns in TEO_c were able to distinguish between true- and false-colored fruits and vegetables (e.g., green kiwi versus red kiwi). We also revealed that TP exhibited different patterns of color memory representation from color patches: it showed stronger responses to true- than false-colored objects; its preference for true-colored objects was diminished in the second half of

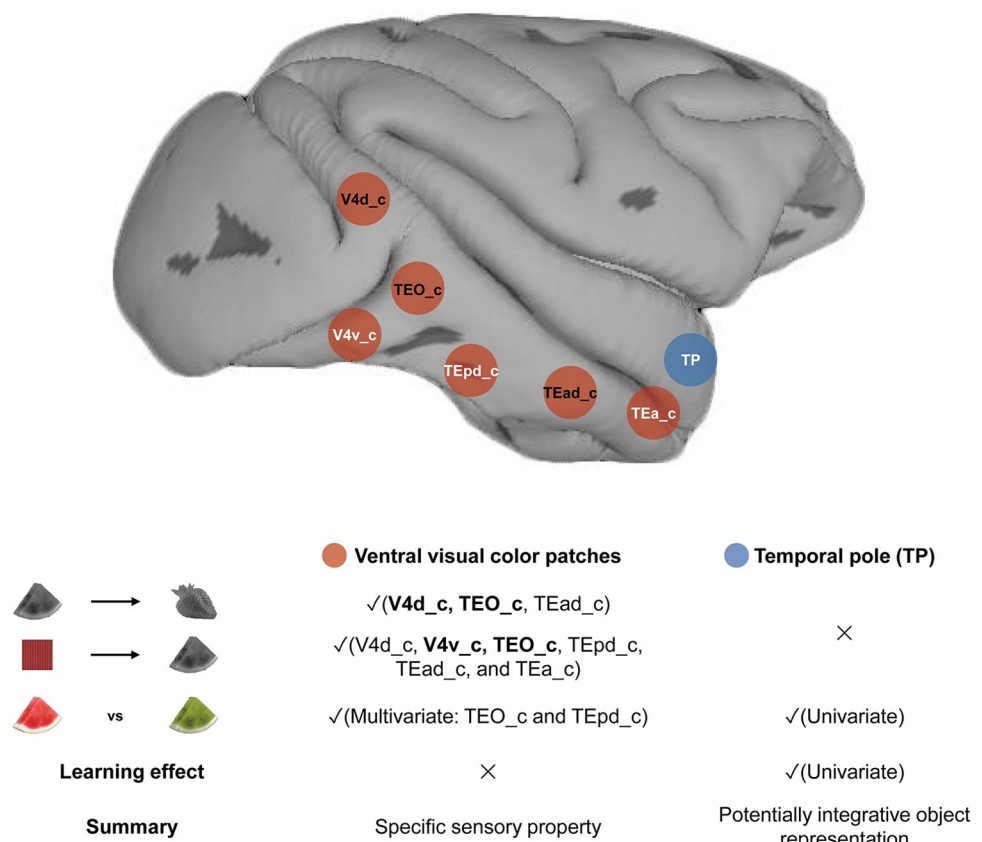

**Fig 4. A schematic view of the distributed object knowledge representation in the macaque brain.** Red nodes: visual color patches that could encode the typical colors of fruits and vegetables presented as grayscale pictures. Red nodes with black characters: color patches (V4d_c, TEO_c, and TEad_c), where brain activity patterns could be transferrable among grayscale objects within the same diagnostic color category. The brain regions highlighted in bold in the table indicate areas where significant or marginally significant results were consistently observed across all 3 tested monkeys. Additionally, TEO_c could encode true-false color as well, showing stable object color memory representation across 3 analyses. Blue nodes, TP, might store integrative object shape-color information that is abstracted away from specific sensory properties, which showed stronger responses to true- than false-colored objects and could not decode memory colors of grayscale objects. Only TP but not the above-mentioned color patches showed a short-term learning effect on false-colored objects. TP, temporal pole.

experimental sessions, indicating its sensitivity to short-term learning; however, it could not encode memory colors of grayscale object images (Fig 4).

The most novel finding of our study is the shared neural representation for chromatic grating perception and object color knowledge in the macaque brain. Our chromatic and achromatic grating viewing experiment identified color patches that responded more strongly to chromatic gratings than achromatic gratings, consistent with the literature before in macaques [37]. In addition, we found that these color patches were capable of distinguishing among red, green, and yellow. Furthermore, we demonstrated that color patches in V4d_c, TEO_c, and TEad_c could encode grayscale strawberries in red and grayscale kiwis in green, i.e., representing the color knowledge based on past perceptual experiences, just like the human brain does [11,41]. Such object color knowledge representation could not solely be attributable to shape confound: (1) brain activities to grayscale images of objects could be classified based on color beyond potential shape similarity; and (2) most compellingly, classifiers trained by chromatic grating viewing where shape information was fully uninformative could successfully decode grayscale objects based on their experienced color. Note that there is one possibility that shape

information could still potentially contribute to the decoding of grayscale objects with different memory colors if there is a certain association between color and shape prevalently present in the macaques. Such a possibility has seldom been investigated in macaques at least to our knowledge (but see [52] and [36] for the possible association between face and warm hue]. In human studies, certain color-shape associations have been proposed (e.g., red-square, yellow-triangle, blue-circle) but with messily inconsistent empirical results [53,54]. Given the lack of evidence for specific color-shape associations in macaques and the fact that the proposed (but not proven) color-shape association in humans could only interfere with our memory color decoding, our current findings indicate that, without actual color inputs, the macaque brain can fill in the colors of a grayscale object based on past color perceptual experiences, a format of "semantic memory" in monkeys.

While some color patches (i.e., V4d_c, TEO_c, and TEad_c) showed robust evidence for object color knowledge across multiple memory color decoding analyses, some showed variations across different analyses. Specifically, color patches in V4v_c, TEpd_c, and TEa_c showed significant results in the cross-decoding analyses between chromatic gratings and grayscale object images, but not in the cross-decoding analyses between different grayscale images. This might be because the latter analyses were performed by split-half analyses with half of the item samples in the former ones and/or because in the latter ones the classifier also picked up non-color information in the object images, potentially impacting the classifier's performance. In addition, in Exp 3, only a limited number of color patches displayed discriminability between true and false conditions based on their activation patterns. This finding could potentially be attributed to the impact of the processing of visual perceptual information, which was intentionally perfectly matched between the true- and false-colored conditions in our study, in some of the color patches. Previous magnetoencephalography (MEG) experiments in humans have shown the interplay between the processing of incoming color information and stored object knowledge [55]. Moreover, the variable patterns of decoding results between Exp 2 and Exp 3 in some color patches (e.g., TEad_c and TEpd_c) could potentially be explained by the visual color processing characteristics in each region [36]. The current experiments were not optimally designed to test such possibilities. Future research, such as studies using methods with better time resolution like electrophysiological recording, may offer valuable insights into the relationship between visual perception and memory representation in the color patches.

We also investigated the memory color encoding in V1 and found that V1 could successfully differentiate memory colors based on chromatic grating training but not when trained on grayscale objects. Note that in previous human studies, memory color representation in V1 has been inconsistently reported [10,11,41].

Previous studies in monkeys have observed backward signal transportation from ATL (e.g., PR) to TE during the visual pair-association task [56] and revealed that the laminar-specific top-down regulation from PR to TE is coupled with a successful recall of visual associates from memory [57], suggesting the engagement of the visual cortex in visual memory retrieval. However, the precise role of the visual cortex is not clear. Our finding clearly revealed the storage of object color memory in color patches. Could it be possible that what we observed in color patches is only a byproduct, a copy transported from object color memory stored elsewhere? A set of empirical observations argues against this possibility. Firstly, whole-brain searchlight analyses in the current study showed that the color knowledge decoding results for grayscale images were mainly distributed in the color patches, with effects in other brain regions (e.g., the ATL) less apparent. Secondly, human cases with color knowledge specific deficits have been related to having lesions encompassing the homologous territories of color patches in monkeys [13,14], hinting at their potentially necessary role in color knowledge representation, although the

contribution of lesions beyond these regions could not be excluded. Thus, the most parsimonious explanation for our findings is that the sensory-experience-derived neural representation for knowledge constitutes a conservative mechanism for encoding object color memory.

In TP, our finding about stronger responses to object images with familiar color-shape combinations largely resonates with previous macaque findings contrasting familiar versus unfamiliar faces [33]. In addition, our results found that TP exhibited a short-term learning effect, consistent with the conventional finding that the ATL is critical in signaling object familiarity, with its neural responses modulated by repeated stimulus exposure [44–47]. The convergence of the sensitivity to familiar faces and to general shape-color combination knowledge about objects is illuminating. Note that, as shown in S10 and S20 Figs, none of the color patches showed differences between true- and false-colored objects or the short-term learning effect. To scrutinize the possibility that the univariate results in TP may be an emotional effect of familiarity originating from elsewhere in the brain, we conducted analyses in the amygdala, a crucial region involved in emotion processing and directly connected with TP [58,59]. Our results revealed no significant main effect of True-False or interaction effect between Period and True-False in the amygdala (S23 Fig). To further assess whether regions other than the color patches and amygdala showed similar univariate results to those found in TP, we conducted the whole-brain analysis. Though several brain regions showed differences in activation strengths, only TP showed a significant interaction effect between Period (first half versus second half of sessions) and True-False at the group level (S16C Fig). Therefore, it is unlikely that the univariate results observed in TP could be attributed to an emotional response originating from another brain region, such as IT and the amygdala.

In addition to the univariate results, TP also exhibited other different patterns of color memory representation from color patches: without representing specific color attributes, as evidenced by the failure to encode typical colors of grayscale objects. In previous human literature, the ATL was traditionally considered as the site for more integrative semantic memory representations abstracted from specific perceptual features ([7,18,20,26]; but see [60] and [61] for evidence for object color memory representation in dorsal ATL). For example, Coutanche and Thompson-Schill [20] showed that the ATL encodes familiar object identities without further sensitivity to specific object shapes or color properties. Our study revealed the potential encoding of object identity of true-colored stimuli in TP (S24 Fig). Our findings are in accord with the possibility that TP may act as an integration area binding together separate features in monkeys, similar to what has been proposed in human ATL. Note that we did not observe clear evidence in PR. The relatively low signal in PR due to its location could potentially account for these findings. In the previous monkey fMRI study that located familiar face regions in PR [33], a particular device was employed to enhance the signal-to-noise ratio (SNR), which may be necessary to reveal effects in PR. More generally, employing rigorous methodologies and carefully controlling for potential confounding factors, future research may provide more detailed insights into the functional role of the ATL (e.g., TP) in monkeys in relation to potential homologous regions in humans.

We observed an intriguing pattern of empirical results in the true- and false-colored objects viewing experiment: the univariate contrast analysis result did not go along with the multivariate pattern analysis result. Specifically, TP showed univariate activation differences between true- and false- conditions but could not differentiate between them based on their activation patterns. Conversely, some of the color perceptual patches (especially posterior and middle ones) exhibited an opposite functional pattern, discriminating the true versus false conditions only in their spatial activation patterns but not in their mean activation strengths. For TP, it is possible that the multivariate pattern coded the specific identity of a unique concept [19,62], whereas the general conceptual familiarity was only reflected in the neuronal activation

amplitude. Indeed, we observed a high probability of encoding object identity in TP. For color patches, the lack of differentiation in activation strengths between true and false conditions might be due to the perceptual dominance of these regions, as our stimuli were well-matched in their sensory input (e.g., color and shape), whereas the memory-based difference was mainly coded in fine-grained multivariate activation patterns.

There has been limited previous evidence regarding object color knowledge in macaques, with only one study indicating that monkeys could exploit color information for categorization [63]. In this study, a behavioral study demonstrated that monkeys could ignore changes in object shape and generalize from one edible object to another based on color in conjunction with other substance properties. This finding indirectly suggested that macaques may have experiential knowledge of colors. In the present study, we conducted a behavioral experiment to measure whether these animals have developed memory colors for the stimuli we used. Our results showed that all the tested macaques showed a visual preference for true-colored objects, providing direct positive behavioral evidence of object color memory in macaques. Furthermore, our experiments directly reveal that the macaque brain can classify objects based on memory color in the absence of color visual input.

Together, these results paint a picture of distributed object knowledge representation in the macaque brain, with clear evidence about object color memory representations shared with perception in the color patches, and potentially integrative representations in the TP region of ATL. Establishing such neural representations of general world knowledge in macaque has important implications for interpreting human brain data. In humans, language and knowledge representations are notoriously difficult to tease apart [6,22–25]. In fact, congenitally blind individuals, who cannot acquire object color knowledge from sensory experiences, can nonetheless acquire similar semantic structures through the dorsal ATL (dATL) [60,61,64]. Such findings lead to the proposal of a dual-form of knowledge representation in the human brain, including both language-derived and sensory-derived knowledge representations [6]. Since the sensory-experience-independent dATL is intrinsically functionally connected with the visual-experience-derived color knowledge representation in the ventral temporal cortex, potential interactions between these 2 forms of knowledge representations could not be fully excluded in humans. The macaque data critically showed that the knowledge representation could be presented in the perception areas without a developed language system. Note that the potentially different subregions within ATL across species require caution and scrutiny. Whether the language-derived knowledge representation observed in human dATL [6] (i.e., along the dorsal aspect of ATL) finds its (primitive) homologous origin in the macaque brain or is human-specific and evolved with language, remains to be further tested with detailed cross-species anatomical and functional comparisons.

A rich array of neural methodological approaches offered by animal models allows for answering important open questions in the future. For example, the dynamic relationships between the 2 types of object knowledge representations reported in this study, as well as among the multiple patches within each system, which potentially support different levels of knowledge abstraction and integration, are difficult to be examined using fMRI alone. While the lesion evidence in humans provides crucial clues to understanding the causal functionality of the perceptual neural system, it also suffers from the limitations that lesions often involve many structures beyond the regions of interest. In contrast, multiple invasive experimental approaches in nonhuman primate models, such as electrophysiology, tracer, stimulation, and reversible lesion, offer techniques with better spatial and time resolutions and causality inference powers that can be pivotal to resolving these issues [65].

Finally, a few caveats warrant discussion. Previous studies have suggested that color selectivity can be influenced by eccentricity [66,67]. It is plausible that the observed differences

across color patches in our results may be attributed to their eccentricity and the resultant color selectivity. To investigate this possibility, we calculated the color selectivity for the defined color patches (see S3 Text). Our results did not show a clear relationship between the encoding ability for memory color and color selectivity (see S25 Fig). Due to technical and animal health reasons, we were not able to define a retinotopic map for each monkey here. To gain a deeper understanding, future studies could consider acquiring the retinotopic map for each monkey and conducting more precise analyses. Furthermore, the inclusion of an experimental group with visual experience and a control group without relevant experience might be more interesting and more thorough to look at the neural mechanism underlying object memory, especially helpful to provide a more comprehensive understanding of the knowledge-building process. Future studies focusing on the development of memory for object color in juvenile monkeys may provide a more comprehensive understanding of the development of object knowledge.

To conclude, we depicted how the macaque brain stores color knowledge about objects in the world: object color knowledge is stored based on visual perceptual neural activities. These findings indicate that the sensory-derived knowledge representation previously observed in the human brain is conservative and not the results of language, and thus open a new avenue for studying various aspects of semantic memory neural mechanisms with nonhuman primate models.

## Methods

### Subjects and general procedures

Three male rhesus monkeys (M1-M3; *Macaca mulatta*; 9 to 10 y old; 7.5 to 10 kg) were used. They were acquired from the same primate breeding facility, where they had social group histories as well as group-housing experiences until their transfer to the Institute of Biophysics (IBP), Chinese Academy of Sciences (CAS) at the age of approximately 4 years. After that, they were individually caged with auditory and visual contact with other conspecifics in the same colony room, which accommodates about 10 rhesus monkeys. All animals used in this study had been housed at IBP for 4 to 6 years before this experiment. All experimental procedures complied with the US National Institutes of Health Guide for the Care and Use of Laboratory Animals and were approved by the Institutional Animal Care and Use Committee of IBP (IBP-NHP-003). Each monkey was surgically implanted with a magnetic resonance (MR)-compatible head post under sterile conditions, using isoflurane anesthesia. After recovery, subjects were trained to sit in a plastic restraint chair and fixate on a central target for long durations with their heads fixed, facing a screen on which visual stimuli were presented [68,69].

### Brain activity measurements

Functional and anatomical MRI scanning was carried out in the Beijing MRI Center for Brain Research (BMCBR). Before each scanning session, an exogenous contrast agent [monocrystalline iron oxide nanocolloid (MION)] was injected into the femoral or external saphenous vein (8 mg/kg) to increase the contrast/noise ratio and to optimize the localization of fMRI signals [70]. Imaging data were collected in a 3T Siemens Prisma MRI scanner with a surface coil array (8 elements). Forty-eight 1.5-mm coronal slices (no gap) were acquired using single-shot interleaved gradient-recalled echo planar imaging. Imaging parameters were as follows: voxel size: 1.5 mm isotropic, field of view: $129 \times 129$ mm; matrix size: $86 \times 86$; echo time (TE): 17 ms; repetition time (TR): 2.5 s; flip angle: 90˚. A low-resolution T2 anatomical scan was also acquired in each session to serve as an anatomical reference (0.625 mm $\times$ 0.625 mm $\times$ 1.5mm; TE: 101 ms; TR: 11.2 s; flip angle: 126˚). To facilitate the alignment to the template, we also

acquired high-resolution T1-weighted whole-brain anatomical scans in separate sessions. Imaging parameters were as follows: voxel size = 0.5 mm isotropic, TE = 2.84 ms, TR = 2.2 s, flip angle: 8°.

## Stimuli

We carefully selected 6 types of color-diagnostic objects of 3 color categories (Red: strawberry, watermelon; Green: cabbage, kiwi; Yellow: banana, corn) as stimuli, adhering to the following principles:

1. We chose fruits and vegetables of one diagnostic color based on the monkeys' prior experiences. For example, apples were excluded since monkeys in our facility have encountered and consumed red, green, and yellow varieties. Conversely, cabbage, despite the potential for different colors such as green and purple, was chosen for the green category, which was the only one provided to the animals in our facility.

2. All the monkeys used in these experiments had similar previous experiences with the selected fruits and vegetables. Monkeys have been exposed to the fruits and vegetables used in the present study for 5 to 6 years, with similar frequency each year, to ensure their nutritional needs and rich environments.

In addition, during the whole study, all monkeys were fed daily the 6 kinds of fruits and vegetables used in the experiments: strawberry, watermelon, cabbage, kiwi, banana, and corn.

Color images of the aforementioned fruits and vegetables were obtained from the internet. To minimize the potential confounding effects of low-level features on the following analyses, we carefully selected and preprocessed our stimuli. Firstly, we matched the size (the number of pixels, S6B Fig) of foreground stimuli. As for shape, we balanced the similarity of shapes for the red and green color-diagnostic categories by ensuring that the shape similarity within the same color category was not significantly different from the similarity between color categories (S6A Fig). However, aside from bananas and corn, identifying other fruits or vegetables with exclusively yellow color and which monkeys encounter as frequently as other objects used in the present study posed difficulties. Therefore, despite having a dissimilar shape in comparison to the objects in the other 2 color categories (S6A Fig), bananas and corn were still selected in the present study.

**Shape similarity calculation.** To quantify the shape dissimilarity between objects, we extracted the object shape boundaries for each image. Subsequently, we computed the Procrustes distance [71], which is a widely accepted method for calculating shape dissimilarity. This distance measure enables the translation, reflection, and orthogonal rotation of the shape matrix, facilitating the calculation of the minimum distance between 2 shapes by minimizing the effect of the presentation position and angle of the objects. Finally, we obtained the average shape similarity by averaging the shape similarity (1 minus the shape dissimilarity) across all pairs of images for each pair of objects.

**Color similarity calculation.** We also evaluated the real color similarity among the 6 types of objects. Specifically, we first extracted the average RGB values from the original color images. Subsequently, we calculated the color similarity between pairs of images by employing the Euclidean distance measurement, as described by Eq 1. Lastly, we computed the average color similarity by taking the mean across all pairs of images for each object pair.

$$Color\ Similarity = 1 - \frac{\sqrt{(R1 - R2)^2 + (G1 - G2)^2 + (B1 - B2)^2}}{\sqrt{(255 - 0)^2 + (255 - 0)^2 + (255 - 0)^2}} \qquad \text{Eq 1}$$

Where R1, G1, and B1 represent the red, green, and blue values of Image 1, respectively. Similarly, R2, G2, and B2 represent the red, green, and blue values of Image 2, respectively.

### Experimental design and task

Experiments were conducted with MATLAB 2016a (MathWorks, Natick, Massachusetts, United States of America) and PsychToolbox-3 [72]. The stimuli were projected on a translucent screen with an MRI-compatible projector at 1024 × 768 at 60 Hz resolution. The viewing distance for monkeys was 68 cm.

In all the experiments, a white fixation cross (0.3˚) was presented at the center of the screen throughout the entire run. Monkeys were required to maintain their gaze within approximately 2.5˚ of the central fixation cross to receive a liquid reward. Eye position was monitored with an infrared pupil tracking system (ISCAN). In the reward schedule, the reward frequency increases as the fixation duration increases [68,69]. Only data in which fixation was maintained well were included in the final analyses [Mean (SE) = 88.80% (0.45%)].

**Chromatic and achromatic grating viewing (Exp 1).** We designed 3 equiluminant (approximately 14.55 cd/m$^2$) chromatic gratings (red, green, and yellow) calibrated by a spectroradiometer (PR655, Spectrascan, Chatsworth, California, USA) at the maximum saturation allowed by the projector. Then, 3 achromatic gratings (luminance contrast 25%, 50%, and 75%) with equal mean luminance to chromatic gratings were defined [37,38,73] (see S5 Table for more details of gratings). Grating stimuli were presented as vertical trapezoid-wave type [11.58 degrees of visual angle (dva) × 11.58 dva]. Each chromatic and achromatic grating drifted back and forth for 15 s, switching directions every 3 s (2.9 cycles/˚, drifting 0.75 cycles/ s). Grating blocks were interleaved with baseline blocks, in which an equiluminant neutral full-field gray (approximately 14.55 cd/m$^2$) background was presented for 10 s. Individual runs began and ended with a baseline block. Different pseudorandom sequences were used in each run. Each monkey was scanned in 2 to 4 sessions, resulting in a total of 20 to 34 runs to use in the final analyses (M1: 20 runs, M2: 34 runs, M3: 25 runs).

**Grayscale object viewing (Exp 2).** Six types of color-diagnostic grayscale objects from 3 color categories (Red: strawberry, watermelon; Green: cabbage, kiwi; Yellow: banana, corn) were presented in separate blocks. The average dva of stimuli was 5.70 (SE = 0.14 degrees) × 6.49 (SE = 0.19 degrees). The images were adjusted to be equiluminant with gratings (CIE L$^*$ = 42.4, approximately 14.55 cd/m$^2$) as described in previous studies [36,41]. Briefly, we converted the colored images to CIE lab color space, adjusted their luminance around the desired value (CIE L$^*$ = 42.4) according to their original SD, then set chromaticity coordinates to 0. We also matched their luminance histograms (S6C Fig). Each object block ($n$ = 6), in which 15 images were each presented for 750 ms followed by a 250 ms interval, lasted 15 s and was presented 3 times in each run. Moreover, each object block alternated with 10 s baseline blocks. Individual runs began and ended with a baseline block. Different pseudorandom sequences were used in each run. Each monkey was scanned in 3 to 4 sessions, resulting in a total of 25 to 32 runs used in the final analyses (M1: 25 runs, M2: 32 runs, M3: 32 runs).

**True- and false-colored object viewing (Exp 3).** In this experiment, we only used stimuli from the red and green color categories (Red: strawberry, watermelon; Green: cabbage, kiwi) to limit the effects of distinct shapes (e.g., the elongated shape of bananas). The average dva of stimuli was 5.51 (SE = 0.10 degrees) × 5.68 (SE = 0.08 degrees). Images of each true-colored object were generated by replacing the pixels in the original achromatic image with a given hue (red strawberry: CIE a$^*$ = 60.00, b$^*$ = 47.00; red watermelon: CIE a$^*$ = 55.00, b$^*$ = 33.00; green cabbage: CIE a$^*$ = −22.25, b$^*$ = 38.27; green kiwi: CIE a$^*$ = −22.64, b$^*$ = 52.76) with preserved luminance of original pixels [52,74]. We then created images of false-colored object

categories (i.e., green strawberry, green watermelon, red cabbage, and red kiwi) by swapping the colors [55]. As such, the overall colors and shapes of true-colored and false-colored categories were fully matched. In the fMRI experiment, there were 8 kinds of object blocks (2 color categories × 2 object categories × true-/false-color). The fMRI design of this experiment was the same as Exp 2. Each monkey was scanned in 2 to 4 sessions, resulting in a total of 20 to 24 runs used in the final analyses (M1: 20 runs, M2: 21 runs, M3: 24 runs).

To avoid interference between experiments caused by the potential learning effect, we conducted the experiments in the following order: Exp 2, Exp 1, and Exp 3. The behavioral experiment was conducted after all fMRI experiments.

## fMRI data analysis

Functional data were preprocessed using Analysis of Functional NeuroImages software (AFNI) [75]. For each session, images were realigned to the volume with the minimum outlier. Then, the data were smoothed with a 2-mm full-width half-maximum Gaussian kernel. Signal intensity was normalized to the mean signal value within each run. We performed a single univariate linear model fit to estimate the response amplitude for each condition. The model included a hemodynamic response predictor for each category and regressors of no interest (baseline, movement parameters from realignment corrections, and signal drifts). A general linear model and an MION kernel were used to model the hemodynamic response function. All fMRI signals throughout the paper have been inverted so that an increase in signal intensity indicates an increase in activation. To identify the anatomical regions for defining ROIs in each monkey, we aligned statistical map images onto images of the symmetric NIMH Macaque Template (NMT) v2 [76]. Then, we projected our statistical results onto a rendered and inflated version of the NMT v2 cortical surface.

## Definition of regions of interest (ROIs)

**Color patches.** First, to obtain the regions sensitive to colors, we compared chromatic grating conditions to the corresponding achromatic grating ones. To further equal the subjective luminance, in accordance with established research in monkeys [37,77], we identified specific chromatic grating conditions that elicited the weakest response in the area MT. Previous studies have demonstrated that responses to moving chromatic gratings in MT, which is specialized to process moving stimuli, can be employed to infer subjective equiluminance [78,79]: MT responds less strongly to moving chromatic gratings if the foreground and background within the chromatic gratings are equiluminant. As such, we identified the chromatic grating conditions that elicited the weakest response in MT, in which the more equiluminant chromatic gratings, the weaker responses evoked [77]. Then, the corresponding achromatic grating conditions were defined as the ones that yielded similar response magnitudes to the selected chromatic grating conditions in V1 [38]. In this way, any region that shows a greater response to equiluminant color than to achromatic luminance can therefore be interpreted as likely making a contribution to hue processing. We identified the color patches for each monkey using the contrast of the above-defined chromatic grating condition versus the corresponding achromatic grating condition (yellow grating versus 50% achromatic grating for M1 and M2; green grating versus 25% achromatic grating for M3). According to previous research [37] and the location of face patches in each monkey, we defined 7 color patches (see S1 Table for the correspondence between names of color patches in the present study and previous studies [37]) that were consistently presented in at least 1 hemisphere across all 3 monkeys at $p < 0.05$ (uncorrected): the color patch around V4d (V4d_c), V4v (V4v_c), TEO (TEO_c), TEpd (TEpd_c), TEad (TEad_c), TEav (TEav_c), and the border between area TEa and IPa (TEa_c)

[76,80,81]. Notably, in 2 monkeys, several color patches in the right hemisphere (i.e., TEpd_c, TEav_c, and TEa_c in M1 and TEad_c in M2) showed relatively weak color bias as in the previous study [37]. We switched the achromatic grating to the next lower level (i.e., from 50% to 25%) to localize them ($p < 0.05$, uncorrected). We defined 3.5-mm radius spheres centered at each cluster mass. To perform the MVPA, the top 50 color-bias voxels within the spheres were selected to yield the final color patches. Given the lack of consistency and small sample size, we refrain from discussing further any left versus right hemispheric differences in the present manuscript, even slight variations were noted. The color patches from both hemispheres (100 voxels) were collapsed respectively in the following ROI analyses and MVPA.

To ensure greater consistency in the locations of the color patches across different monkeys, we conducted a group analysis across the 6 hemispheres from the 3 monkeys (see S2 Text and S12 Fig). Due to the limited number of monkey subjects, we considered each hemisphere as a sample for the group analysis. Despite inter-animal variability, the color patches identified at the individual level were found in close proximity to the color patches identified at the group level.

**True-false ROIs.** As previous studies have revealed the critical role of perirhinal circuits in the ATL in object-associative memory retrieval [26,42], we mainly looked into the 2 key nodes in this circuit: TP and PR. With the contrast of true- and false-colored objects, we observed consistent bilateral activated clusters in all 3 monkeys in TP but not in PR at $p < 0.05$ (uncorrected) (Figs 3A and S13A). In 2 of 3 monkeys, only unilateral activated clusters in PR could be found at $p < 0.05$ (uncorrected), while the contralateral PR clusters only showed at a lenient threshold (left PR in M1: $p < 0.20$, uncorrected; right PR in M2: $p < 0.25$, uncorrected). For the ROI analyses, to avoid the problem of double dipping [82], we split the data yielding 4 combinations and then conducted the ROI analyses [83,84] (S26 Fig). Specifically, we used half of the categories to define true-false voxels with the contrast of true- versus false-colored objects and then conducted the following ROI analyses on the other half of the categories (e.g., defining with true-colored cabbage and strawberry versus false-colored ones, then measuring responses on kiwi and watermelon). For each combination, the top 10 true-false voxels were selected within the anatomical mask of TP/PR from the D99 digital atlas [76,80] to yield each sub-ROI. All 4 combinations were analyzed and then averaged to produce the final results for each monkey. We also defined the ROI for the amygdala in the same way as above. For the MVPA, to ensure the location of ROIs for the univariate analysis and MVPA was close, we collapsed the 4 sub-ROIs and then generated a continuous combined mask, and then selected the top 50 true-false voxels. Note that, for PR, the same definition method as for TP could not yield 50 true-false voxels within PR. We conducted MVPA based on all survived voxels (33 to 35 voxels). To eliminate the impact of the number of voxels on MVPA, we also opted to directly select the top 50 true-false voxels from the anatomical mask of PR, yielding similar results (S27 Fig). To investigate the learning effect in the true- and false-colored objects viewing experiment, we divided the data from the middle into 2 halves (first and second half) for each monkey: M1: 2 versus 2 sessions (12 versus 8 runs); M2: 2 versus 2 sessions (11 versus 10 runs); M3 1 versus 1 session (11 versus 13 runs). Note that the analyses for the true- and false-colored object viewing experiment were conducted when the first and second halves were separated or combined.

## Multivoxel pattern analysis (MVPA)

It is worth noting that the classification analysis is widely utilized in studies focusing on memory and imagination [11,40,55,85–87]. To facilitate meaningful comparisons between our study in monkeys and previous studies involving humans, we employed classification analysis. We applied LDA classifiers to analyze data based on CoSMoMVPA [88]. Prior to analyses,

data were normalized using z-scoring to eliminate the impact of activation intensity to a certain extent [88]. During the chromatic and achromatic grating viewing experiment, we performed three-way classifications for chromatic gratings (i.e., Red, Green, and Yellow) and achromatic gratings (i.e., 25%, 50%, and 75% luminance contrast), respectively. A leave-one-out procedure was conducted. That is, for one iteration, N-1 runs were used for training, and the left-out run was used for testing. To test whether color patches were capable of differentiating between grayscale objects from different diagnostic color categories, we performed the classification of grayscale objects with different memory colors first. Eight-fold cross-validation analyses were conducted: training on half of the grayscale objects (e.g., strawberry, cabbage, and banana) on all runs and testing on the rest half of the grayscale objects (e.g., watermelon slice, kiwi slice, and corn) on each run, in the grayscale object viewing experiment. Accuracy rates from all 8 folds cross-validation analyses were averaged to yield the classification accuracy. To avoid the confounds of object shape, we further excluded objects with yellow diagnostic color (i.e., banana and corn) due to their relatively elongated shapes and did the classification again (4-fold cross-validation). To further verify the memory color representation of grayscale objects from different diagnostic color categories, predicting the color information of grayscale objects, three-way color classifiers (i.e., Red, Green, and Yellow) were trained on activity patterns elicited by the chromatic grating stimuli and then tested on responses to three grayscale object categories [10,40]. Note that all the runs in the chromatic and achromatic grating viewing experiment were used for training, and testing was conducted for each run in the grayscale object viewing experiment. These memory color decoding methods allowed us to examine the classifier's ability to discriminate based on (memory) color information alone, independent of shape. By employing these methods, we minimized the contribution of low-level features to the decoding results. In the true- and false-colored objects viewing experiment, two-way classifiers (i.e., true- and false-colored objects) were trained. Then, the same leave-one-out procedure as described above was conducted. When decoding object identity, all true-colored objects from the true- and false-colored object viewing experiment were used. Specifically, we trained a four-way classifier (red strawberry, red watermelon, green cabbage, green kiwi) and employed the leave-one-out procedure. Finally, for all MVPA, data from all iterations were averaged, yielding the final classification accuracy.

To investigate whether other brain areas beyond the functionally defined color patches and the true-false ROIs also code color knowledge, we performed whole-brain searchlight analyses (50 voxels for each sphere). Classification methods were the same as those used in the ROI-based decoding analyses.

## Statistical analysis

All analyses were conducted using SPSS (v25) software (SPSS, Chicago, Illinois, USA), Matlab 2016a (MathWorks, Natick, Massachusetts, United States) and "R" statistical programming language (R Foundation for Statistical Computing, R Development Core Team, 2017). We perform GLMMs to compare responses to chromatic and achromatic grating in color patches, with Monkey, Session, and Run as random factors [89,90]. Notably, to avoid the double-dipping problem, we performed the ROI analyses based on the responses to the gratings that were not used to define the color patches. When comparing the responses to 3 categories of grayscale objects in color patches and true-false ROIs, we also employed GLMM. Similarly, GLMMs were conducted to compare the responses to true- and false-colored objects in all the defined ROIs. To further explore the potential effects of learning, GLMMs were also conducted with Period (first half and second half of sessions) and True-False as fixed factors with Monkey, Session, and Run as random factors.

For the results of MVPA, to limit the possibility of result bias by a single monkey, we also performed GLMMs to compare decoding accuracy with the chance level, with Monkey, Session, and Run as random factors. Considering the binomial distribution nature of decoding accuracy [91,92], we employed a binomial target distribution with a log link function in the present study.

FDR corrections were conducted to adjust multiple ROIs ($n$ = 7 for all the analyses for color patches, $n$ = 2 for all the analyses for ATL) [33].

The binomial tests were conducted for the searchlight decoding analyses [85,93], in which accuracy was estimated on the total number of correctly classified samples given the overall number of cross-validation folds.

## Supporting information

**S1 Fig. Experimental procedures and results of the behavioral experiment.** (A) The trial procedure: fixation, free viewing, and reward after successful trial completion. (B) From left to right, the proportion of fixation time averaged across 2 monkeys and for each monkey. (C) From left to right, the proportion of first fixation averaged across 2 monkeys and for each monkey. Bars display mean values +/− SEM. Red asterisks indicate a significant difference between responses evoked by true- and false-colored stimuli in B and C; **$p$ < 0.01, ***$p$ < 0.001. The data underlying this figure are available in S1 Data.
(PDF)

**S2 Fig. FMRI responses and decoding results in color patches in the chromatic and achromatic gratings viewing experiment.** (A) Averaged fMRI responses elicited by chromatic and achromatic gratings, which were not used to define color patches, in color patches across all 3 subjects. (B) Chromatic and achromatic decoding accuracy when training the classifier to distinguish among the 3 chromatic/achromatic gratings in N-1 runs and testing on the left-out run. Black asterisks indicate a significant difference from the chance level (0.333 in B, indicated by the dash lines), and red ones indicate a significant difference between chromatic and achromatic gratings; *q < 0.05, **q < 0.01, ***q < 0.001. The data underlying this figure are available in S1 Data.
(PDF)

**S3 Fig. Decoding results in color patches in the chromatic and achromatic gratings viewing experiment in each subject.** Bars display mean values +/− SEM. Black asterisks indicate a significant difference from the chance level (0.333, indicated by the dash lines), and red ones indicate a significant difference between chromatic and achromatic gratings; *$p$ < 0.05, **$p$ < 0.01, ***$p$ < 0.001. The numbers above the bars indicate $p$-values that are marginally significant ($p$ < 0.1). The data underlying this figure are available in S1 Data.
(PDF)

**S4 Fig. FMRI responses to 3 color-diagnostic grayscale objects.** (A–C) Averaged fMRI responses to 3 categories of color-diagnostic grayscale objects in color patches defined on individual activation maps (A) and by the group analysis (B) and in ATL (C) across all 3 monkeys. Bars display mean values +/− SEM. The data underlying this figure are available in S1 Data.
(PDF)

**S5 Fig. Classification of grayscale objects with different memory colors in color patches excluding shape confound.** (A) The results of classification of grayscale objects with 3 different memory colors: training the classifier to distinguish among the half of grayscale objects (e.g., strawberry, cabbage, and banana) and testing on the other half (e.g., watermelon slice,

kiwi slice, and corn). Successful decoding was found in V4d_c, V4v_c, TEO_c, TEad_c, and TEa_c. (B) The "memory" color and shape similarities between training and test objects with red and green memory colors in various fold combinations. The second rows exhibited non-simultaneous alterations in "memory" color and shape similarity matrix values (framed by the red square), which were used to compute decoding accuracy rates in (C). For example, in Fold 1, we trained the classifier to distinguish the grayscale images of cabbage and watermelon and tested on kiwi and strawberry. In this case, kiwi might be correctly classified along with cabbage based on either color or shape, as for both properties, kiwi is more similar to cabbage than to watermelon. By contrast, strawberry is closer to watermelon in color but to cabbage in shape. Therefore, if a classifier identifies strawberries as more similar to watermelons with an accuracy significantly higher than chance, such a classifier could not be based on shape. (C) The results of the classification of grayscale objects with red and green memory colors in color patches based on the second rows of folds. Successful decoding was found in V4d_c, TEO_c, and TEad_c after excluding shape confound. Bars display mean values +/− SEM. Black asterisks indicate a significant difference from the chance level (0.333 in A, 0.5 in C, indicated by the dashed lines); *$q < 0.05$, **$q < 0.01$, ***$q < 0.001$. The data underlying this figure are available in S1 Data.
(PDF)

**S6 Fig. Statistical information of low-level stimulus features.** (A) The averaged shape similarity matrix of grayscale images from different color categories. (B) Pixel counts of foreground of grayscale images from different color categories. Bars display mean values +/− SEM. (C) Luminance histograms of foreground of grayscale images from different color categories. The data underlying this figure are available in S1 Data.
(PDF)

**S7 Fig. Classification of grayscale objects with red and green memory colors in each subject.** (A–C) Classification of grayscale objects with red and green memory colors in each monkey. Bars display mean values +/− SEM. Black asterisks indicate a significant difference from the chance level (0.5, indicated by the dashed lines); *$p < 0.05$, **$p < 0.01$, ***$p < 0.001$. (D–F) The results of whole-brain searchlight analyses for decoding memory color in each subject shown on the template inflated surface. White solid lines indicate color patches and TP defined for each subject. The data underlying this figure are available in S1 Data.
(PDF)

**S8 Fig. Memory color decoding and searchlight results based on chromatic gratings training in each subject.** (A–C) Memory color decoding accuracy in color patches in each monkey. Bars display mean values +/− SEM. Black asterisks indicate a significant difference from the chance level (0.333, indicated by the dashed lines); *$p < 0.05$, **$p < 0.01$, ***$p < 0.001$. The numbers above the bars indicate $p$-values that are marginally significant ($p < 0.1$). (D–F) The results of whole-brain searchlight analyses for decoding memory color in each subject shown on the template inflated surface. White solid lines indicate color patches and TP defined for each subject. The data underlying this figure are available in S1 Data.
(PDF)

**S9 Fig. Results of memory color decoding in V1.** (A) Results of classification of grayscale objects with red and green memory colors: training the classifier to distinguish half set of the red and green color-diagnostic grayscale objects and testing on the other half in Exp 2. (B) Results of memory color decoding based on chromatic gratings training: training the classifier to distinguish among 3 chromatic gratings in Exp 1 and then testing on 3 categories of grayscale objects in Exp 2. Bars display mean values +/− SEM. Black asterisks indicate a significant

difference from the chance level (0.5 in A, 0.333 in B, indicated by the dashed lines); $^*p < 0.05$. The data underlying this figure are available in S1 Data.
(PDF)

**S10 Fig. Averaged fMRI responses to true- and false-colored objects in color patches across 3 monkeys.** No significant differences were found between responses evoked by true- and false-color objects. Bars display mean values +/− SEM. The data underlying this figure are available in S1 Data.
(PDF)

**S11 Fig. True-False color decoding and searchlight results in each subject.** (A–C) True-False color decoding accuracy in color patches in each monkey. Bars display mean values +/− SEM. Black asterisks indicate a significant difference from the chance level (0.5, indicated by the dashed lines); $^*p < 0.05$, $^{**}p < 0.01$, $^{***}p < 0.001$. (D–F) The results of whole-brain searchlight analyses for decoding true-false color for each subject shown on the template inflated surface. White solid lines indicate color patches and TP defined for each subject. The data underlying this figure are available in S1 Data.
(PDF)

**S12 Fig. Results of color patches defined by the group analysis.** (A) Color biased clusters defined by the group analysis based on the 6 hemispheres from the 3 subjects ($p < 0.01$, uncorrected) from Exp 1 are shown on the lateral view of the template inflated surface of the left hemisphere. Red solid lines indicate that clusters could be defined at $p < 0.05$ (uncorrected), while white solid lines indicate clusters that could be defined when adjusting the contrasted achromatic grating to the next lower level (i.e., from 50% to 25% in M1 and M2, $p < 0.01$, uncorrected) to account for the relatively weak color bias in the right hemisphere. Blue bull-seye marks the approximate location of the fovea based on the data set shared by Janssens and colleagues [94], which was downloaded from https://gbiomed.kuleuven.be/english/research/50000666/50000669/50488669/neuroserv/publications/JNEUROSCI. The permission to use dataset from Janssens and colleagues [94](https://www.jneurosci.org/content/34/31/10156) has been granted by the corresponding author, Wim Vanduffel. (B) Averaged fMRI responses elicited by chromatic and achromatic gratings, which were not used to define color patches, in color patches defined by the group analysis across 3 monkeys. Red asterisks indicate a significant difference between chromatic and achromatic gratings; $^*q < 0.05$, $^{***}q < 0.001$. (C) Chromatic and achromatic decoding accuracy when training the classifier to distinguish among the 3 chromatic/achromatic gratings in N-1 runs and testing on the left-out run in Exp 1. (D) Results of classification of grayscale objects with red and green memory colors: training the classifier to distinguish half set of the red and green color-diagnostic grayscale objects and testing on the other half in Exp 2. (E) Results of memory color decoding based on chromatic gratings training: training the classifier to distinguish among 3 chromatic gratings in Exp 1 and then testing on 3 categories of grayscale objects in Exp 2. (F) Results of true-false color decoding: training on true- and false-colored objects in N-1 runs and testing on the left-out run in Exp 3. Bars display mean values +/− SEM. Black asterisks indicate a significant difference from the chance level (0.333 in C and E, 0.5 in D and F, indicated by the dashed lines); $^*q < 0.05$, $^{**}q < 0.01$, $^{***}q < 0.001$. The data underlying this figure are available in S1 Data.
(PDF)

**S13 Fig. FMRI responses to true- and false-colored objects in PR.** (A) The true-color versus false-color in PR are shown in the coronal slices for each of the 3 subjects (M1 to M3). The threshold for each cluster is listed on the right side of the slices. (B, C) The true-color versus false-color in the first part of sessions (B), and the second part of sessions (C) in PR are shown

in the coronal slices for each of the 3 subjects (M1 to M3) at $p < 0.05$, respectively. Each slice's anterior/posterior position is indicated on the top left corner (mm relative to the interaural canal). (D) Averaged fMRI responses to true- and false-colored objects in PR across all 3 subjects based on all the sessions [$F_{(1,128)} = 0.050$, $q = 0.823$, $\eta2 < 0.001$; two-tailed]. (E) Averaged fMRI responses to true- and false-colored objects in PR across all 3 subjects from the first and second halves of sessions [the interaction effect between Period (first half versus second half) and True-False across 3 monkeys: $F_{(1,126)} = 0.305$, $q = 0.582$, $\eta2 = 0.002$]. The data underlying this figure are available in S1 Data.
(PDF)

**S14 Fig. Regions exhibiting stronger responses to false-colored stimuli compared to true-colored stimuli.** Results of the whole-brain analysis at the group level are shown on the lateral view of the template inflated surface.
(PDF)

**S15 Fig. The interaction effect in TP for each subject.** (A) The conjunction map that combined the true-color versus false-color contrast with the interaction effect between Period (first half versus second half of sessions) and True-False for each of the 3 subjects (M1 to M3) at $p < 0.01$. Each slice's anterior/posterior position is indicated on the top left corner (mm relative to the interaural canal). (B–D) FMRI responses to true- and false-colored objects in TP from the first and second halves of sessions for each subject, respectively. Individual analyses demonstrated similar trends for the 3 monkeys, although the results were not significant, likely due to the limited sample size used as a result of the request to avoid double dipping and the impact of the learning effect. (E–G) fMRI responses to true- and false-colored objects in TP for each session in each subject. Bars display mean values +/− SEM. Red asterisks indicate a significant difference between responses evoked by true- and false-colored stimuli; $^*p < 0.05$, $^{**}p < 0.01$, $^{***}p < 0.001$. The data underlying this figure are available in S1 Data.
(PDF)

**S16 Fig. TP exhibiting the interaction effect at the group level.** (A) Averaged fMRI responses to true- and false-colored objects in TP across all sessions and subjects. To avoid the possible circularity, we used a one-half set of the true- and false-colored objects to define the true-false ROI in TP and then examined the true-false differentiation effect on the other half set of the stimuli. Due to this strict approach, differences between true- and false-colored objects when combining all sessions were not visible [$F_{(1,128)} = 0.274$, $q = 0.823$, $\eta2 = 0.002$; two-tailed]. (B) Averaged fMRI responses to true- and false-colored objects in TP across all 3 subjects from the first and second halves of sessions. We did observe an interesting learning effect: a significant interaction effect between Period and True-False across 3 monkeys [$F_{(1,126)} = 11.006$, $q = 0.002$, $\eta2 = 0.080$; two-tailed]. To further investigate this interaction effect, we conducted post hoc analyses. In the first half of the sessions, the ROI analysis revealed that TP responded significantly more strongly to true-colored objects across 3 monkeys [$t_{(126)} = 3.420$, $q = 0.002$, Cohen's $d = 0.609$; two-tailed]. The differences in averaged responses evoked by true- and false-colored objects across all 3 monkeys observed in the first half of the sessions vanished in the second half. Bars display mean values +/− SEM. Red asterisks indicate a significant difference between responses evoked by true- and false-colored stimuli; $^{**}q < 0.01$. (C) Regions exhibiting the interaction effect between Period (first half versus second half of sessions) and True-False of the whole-brain analysis at the group level are shown on the lateral view of the template inflated surface. White solid lines indicate the location of TP from the D99 atlas [76,80]. The data underlying this figure are available in S1 Data.
(PDF)

**S17 Fig. FMRI responses to true- and false-colored objects in PR from the first and second halves of sessions for each subject.** (A–C) FMRI responses to true- and false-colored objects in PR from the first and second halves of sessions for each subject, respectively. Bars display mean values +/− SEM. Red asterisks indicate a significant difference between responses evoked by true- and false-colored stimuli; **$p < 0.01$. The numbers above the bars indicate $p$-values that are marginally significant ($p < 0.1$). The data underlying this figure are available in S1 Data.
(PDF)

**S18 Fig. True-false color decoding accuracy in true-false ROIs for each subject.** (A–C) True-False color decoding accuracies in TP and PR in each monkey when combining all sessions, respectively. (D, E) True-false color decoding accuracies in TP and PR in each monkey based on the first and second halves of sessions, respectively. Bars display mean values +/− SEM. Black asterisks indicate a significant difference from the chance level (0.5, indicated by the dashed lines); *$p < 0.05$, **$p < 0.01$. The numbers above the bars indicate $p$-values that are marginally significant ($p < 0.1$). The data underlying this figure are available in S1 Data.
(PDF)

**S19 Fig. Averaged true-false color decoding accuracy in true-false ROIs across 3 monkeys.** (A) True-False color decoding accuracies in TP and PR across 3 monkeys based on the first and second halves of sessions. (B) True-False color decoding accuracies across 3 monkeys when combining all sessions. Bars display mean values +/− SEM. Black asterisks indicate a significant difference from the chance level (0.5, indicated by the dashed lines); *$q < 0.05$. The data underlying this figure are available in S1 Data.
(PDF)

**S20 Fig. Averaged fMRI responses and true-false color decoding in color patches based on the first and second halves of sessions.** (A) Averaged fMRI responses to true- and false-colored objects in color patches across all three subjects for the first and second halves of sessions. (B) True-false color decoding accuracies across 3 monkeys for the first and second halves of sessions. Bars display mean values +/− SEM. Black asterisks indicate a significant difference from the chance level (0.5, indicated by the dashed lines); *$q < 0.05$. The data underlying this figure are available in S1 Data.
(PDF)

**S21 Fig. True-false color decoding accuracies in each monkey for the first and second halves of sessions.** Bars display mean values +/− SEM. Black asterisks indicate a significant difference from the chance level (0.5, indicated by the dashed lines). Red asterisks indicate a significant difference between the first and second half of sessions; *$p < 0.05$. The numbers above the bars indicate $p$-values that are marginally significant ($p < 0.1$). The data underlying this figure are available in S1 Data.
(PDF)

**S22 Fig. Memory color decoding results in true-false ROIs for each monkey.** (A) Results of classification of grayscale objects with red and green memory colors in each monkey: training the classifier to distinguish half set of the red and green color-diagnostic grayscale objects and testing on the other half in Exp 2. (B) Results of memory color decoding based on chromatic gratings training in each monkey: training the classifier to distinguish among three chromatic gratings in Exp 1 and then testing on 3 categories of grayscale objects in Exp 2. Bars display mean values +/− SEM. Dashed lines indicate the chance level (0.5 in A and 0.333 in B); *$p < 0.05$. The numbers above the bars indicate $p$-values that are marginally significant

($p < 0.1$). The data underlying this figure are available in S1 Data.
(PDF)

**S23 Fig. Averaged fMRI responses to true- and false-colored objects in the amygdala across all 3 subjects from the first and second halves of sessions.** The main effect of true-false [$F_{(1,126)} = 2.348$, $p = 0.128$] and the interaction effect between Period and true-false [$F_{(1,126)} = 0.733$, $p = 0.393$] were not significant. Bars display mean values +/− SEM. n.s., not significant. The data underlying this figure are available in S1 Data.
(PDF)

**S24 Fig. Results of object identity encoding in TP and PR.** (A) Object identity encoding in TP and PR across 3 monkeys utilizing all true-colored objects in Exp 3. The group analysis yielded a trend but not significant decoding accuracy above the chance level for TP [$F_{(1,128)} = 2.281$, $p = 0.067$, q = 0.126, one-tailed]. (B–D) Object identity encoding in TP and PR in each monkey, respectively. Bars display mean values +/− SEM. Dashed lines indicate the chance level (0.25). The numbers above the bars indicate $p$-values that are marginally significant ($p < 0.1$). The data underlying this figure are available in S1 Data.
(PDF)

**S25 Fig. Color selectivity in color patches.** (A) Color selectivity in color patches defined by the individual analysis. (B) Correlations between the color selectivity and decoding accuracies of classification of grayscale objects with red and green memory colors (two-tailed Spearman correlation, r = 0.595, $p = 0.170$) and memory color decoding based on chromatic gratings training (two-tailed Spearman correlation, r = 0.179, $p = 0.713$) in color patches defined on the individual monkey's activation map. (C) Color selectivity in color patches defined by the group analysis. (D) Correlations between the color selectivity and decoding accuracies of classification of grayscale objects with red and green memory colors (two-tailed Spearman correlation, r = −0.214, $p = 0.662$) and memory color decoding based on chromatic gratings training (two-tailed Spearman correlation, r = 0.214, $p = 0.662$) and in color patches defined by the group analysis. +q $< 0.1$, *q $< 0.05$, **q $< 0.01$, ***q $< 0.001$, Bonferroni corrected. The data underlying this figure are available in S1 Data.
(PDF)

**S26 Fig. The 4 combinations used for defining true-false ROIs.** Half of the objects were used to define true-false ROIs in the ATL with the contrast of true-colored objects versus false-colored objects and then conducted the ROI analyses on the other half of the objects (e.g., in Combination 1, defining ROIs with true-colored cabbage and strawberry versus false-colored ones, then measuring responses on true- and false-colored kiwi and watermelon).
(PDF)

**S27 Fig. Averaged decoding accuracy in PR defined by directly selecting the top 50 true-false voxels from the anatomical mask of PR from the D99 digital atlas across all 3 subjects.** (A) Results of classification of grayscale objects with red and green memory colors: training the classifier to distinguish half set of the red and green color-diagnostic grayscale objects and testing on the other half in Exp 2. (B) Results of memory color decoding based on chromatic gratings training: training the classifier to distinguish among 3 chromatic gratings in Exp 1 and then testing on 3 categories of grayscale objects in Exp 2. (C) Results of true-false color decoding: training on true- and false-colored objects in N-1 runs and testing on the left-out run in Exp 3 when combining all sessions, in the first half of sessions and in the second half of sessions. (D) Results of object identity decoding: training on all true-colored objects in N-1 runs and testing on the left-out run when combining all sessions in Exp 3. Bars display mean

values +/− SEM. Dashed lines indicate the chance level (0.5 in A and C; 0.333 in B; 0.25 in D); *q < 0.05. The data underlying this figure are available in S1 Data.
(PDF)

**S1 Table. The correspondence between names of color patches in the present study and previous studies [37].**
(DOCX)

**S2 Table. GLMM results of main effects of true-false in color patches.**
(DOCX)

**S3 Table. GLMM results of fMRI responses to true- and false-colored objects in TP and PR for each subject.**
(DOCX)

**S4 Table. GLMM results of interaction effects between Period and true-false across 3 monkeys in color patches.**
(DOCX)

**S5 Table. Parameters of chromatic and achromatic stimuli used in Exp 1.**
(DOCX)

**S1 Text. Experimental procedures and data analysis for behavioral experiment.**
(DOCX)

**S2 Text. Definition of color patches by the group analysis and the corresponding results.**
(DOCX)

**S3 Text. Calculation of color selectivity index.**
(DOCX)

**S1 Data. Data underlying the plots in all figures.**
(XLSX)

## Acknowledgments

We thank Tianshu Yang and Zhaojin Cheng for the animal surgery.

## Author Contributions

**Conceptualization:** Minghui Zhao, Xiaoying Wang, Yanchao Bi, Ning Liu.

**Data curation:** Minghui Zhao, Ning Liu.

**Formal analysis:** Minghui Zhao, Ning Liu.

**Funding acquisition:** Xiaoying Wang, Yanchao Bi, Ning Liu.

**Investigation:** Minghui Zhao, Yumeng Xin, Haoyun Deng, Zhentao Zuo, Ning Liu.

**Methodology:** Minghui Zhao, Yumeng Xin, Xiaoying Wang, Yanchao Bi, Ning Liu.

**Project administration:** Xiaoying Wang, Yanchao Bi, Ning Liu.

**Resources:** Zhentao Zuo, Ning Liu.

**Software:** Minghui Zhao, Yumeng Xin, Ning Liu.

**Supervision:** Xiaoying Wang, Yanchao Bi, Ning Liu.

**Validation:** Minghui Zhao, Ning Liu.

**Visualization:** Minghui Zhao.

**Writing – original draft:** Minghui Zhao, Xiaoying Wang, Yanchao Bi.

**Writing – review & editing:** Minghui Zhao, Yumeng Xin, Haoyun Deng, Zhentao Zuo, Xiaoying Wang, Yanchao Bi, Ning Liu.

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
