## [Editor Report · Decision Letter 0]

27 Aug 2024

Dear Dr Liu, 

Thank you for submitting your manuscript entitled "Object color knowledge representation in the macaque brain" for consideration as a Research Article by PLOS Biology.

Your manuscript has now been evaluated by the PLOS Biology editorial staff as well as by an academic editor with relevant expertise and I am writing to let you know that we would like to invite you to fully submit your manuscript, so we can proceed with our editorial checks. After discussing your manuscript and the peer review history with our Academic Editor, we have decided that we are likely to publish your paper without further peer review.

However, before we can continue with our editorial checks, we need you to complete your submission by providing the metadata that is required for full assessment. To this end, please login to Editorial Manager where you will find the paper in the 'Submissions Needing Revisions' folder on your homepage. Please click 'Revise Submission' from the Action Links and complete all additional questions in the submission questionnaire.

Once your full submission is complete, your paper will undergo a series of checks in preparation for publication. After your manuscript has passed the checks it will be sent out for review. To provide the metadata for your submission, please Login to Editorial Manager (https://www.editorialmanager.com/pbiology) within two working days, i.e. by Aug 29 2024 11:59PM.

In your cover letter, please also provide the manuscript ID your manuscript was given at Nature Communications, o we can contact the editor to validate the peer review history. In addition, please upload a response to the reviews as a 'Prior Peer Review' file type, which should include the reports in full and a point-by-point reply detailing how you have addressed the reviewers' concerns. 

Kind regards,

Christian

Christian Schnell, PhD

Senior Editor

PLOS Biology

cschnell@plos.org

---

## [Editor Report · Decision Letter 1]

6 Sep 2024

Dear Dr Liu,

Thank you for your patience while we considered your revised manuscript "Object color knowledge representation in the macaque brain" for publication as a Initial Research Submission at PLOS Biology. This revised version of your manuscript has been evaluated by the PLOS Biology editors and the Academic Editor.

We are likely to accept this manuscript for publication, provided you satisfactorily address the following data and other policy-related requests (please note, however, that I am still waiting to hear from the previous journal to validate the reviewer reports):

* We would like to suggest a different title to improve readability/accuracy: "Object color knowledge representation occurs in the macaque brain despite the absence of a developed language system"

* Please add the links to the funding agencies in the Financial Disclosure statement in the manuscript details

* Please include the full name of the IACUC/ethics committee that reviewed and approved the animal care and use protocol/permit/project license. Please also include an approval number.

* DATA POLICY:

Regardless of the method selected, please ensure that you provide the individual numerical values that underlie the summary data displayed in the following figure panels as they are essential for readers to assess your analysis and to reproduce it: 2CEG, 3DEF, S1BC, S2AB, S3ABC, S4ABC, S5AC, S6B, S7ABC, S8ABC, S9AB, S10, S11ABC, S12BCDEF, S13DE, S15BCDEFG, S16AB, S17ABC, S18ABCDEF, S19AB, S20AB, S21ABC, S22Ab, S23, S24ABCD, S25AB and S27ABCD

* CODE POLICY

We expect to receive your revised manuscript within two weeks. 

*Published Peer Review History*

*Press*

Sincerely,

Christian

Christian Schnell, PhD

Senior Editor

cschnell@plos.org

PLOS Biology

---

## [Editor Report · Decision Letter 2]

21 Sep 2024

Dear Dr Liu,

Thank you for the submission of your revised Initial Research Submission "Object color knowledge representation occurs in the macaque brain despite the absence of a developed language system" for publication in PLOS Biology. On behalf of my colleagues and the Academic Editor, Huan Luo, I am pleased to say that we can in principle accept your manuscript for publication, provided you address any remaining formatting and reporting issues. These will be detailed in an email you should receive within 2-3 business days from our colleagues in the journal operations team; no action is required from you until then. Please note that we will not be able to formally accept your manuscript and schedule it for publication until you have completed any requested changes.

When you attend to the requests to come, could you please also ensure that your code and data are in a citable repository? I saw that those are currently stored at OSF, but this does not seem to have a DOI. This might be because the repository is not yet public. If you have a DOI for at OSF, you can leave it as is. Otherwise, please provide the link to zenodo, as this contains a DOI and makes it easier to cite the code and data you provide. 

PRESS

Sincerely, 

Christian

Christian Schnell, PhD

Senior Editor

PLOS Biology

cschnell@plos.org